# Soft money, hard power: Mapping the material contingencies of change in global health academic structures

Daniel W. Krugman[1]*, Alice Bayingana[2]

1 Department of Anthropology, Brown University, Providence, Rhode Island, United States of America,
2 School of Public Health, University of Sydney, Sydney, New South Wales, Australia

* daniel_krugman@brown.edu

## Abstract

In the proliferating conversations about decolonizing Global Health, a basic assumption has been that global South actors should be the conceptualizers, leaders, and makers of change with global North counterparts thought of as allies or accomplices. This article complicates assumptions about whose responsibility it is to "decolonize" Global Health and how different actors should go about it. We do this through a qualitative anthropological investigation on resource structures and material flows and the ways that global North actors relate to and make sense of them. By outlining the financial structuring at a major American school of public health and faculty experiences within this system, we show how the "soft money" structure reproduces colonial relations, elite dominance, and capture of popularized words connoting change. While acknowledging the necessity of promoting knowledge and discourses from the South, we demonstrate how Global Health academics in this powerful institution hide or overlook their structurally advantageous positions to create change by deploying discourses such as "following the South" or "centering Southern voices." Grappling with how needed material changes fundamentally go against institutional and personal interests of powerful global North institutions and actors, we introduce "ruinous solidarity" as a paradigm of praxis. Ruinous Solidarity is elite actors thinking and acting in ways that embraces the possibilities that emerge in loss of resources and prestige. Ruinous solidarity seeks to move elite actors subjected to funding structures such as those described in this article away from passivity and obscuring the material bases of the imbalance of power in Global Health. Pushing for long term, ethical transfers of wealth and responsibility, ruinous solidarity explicitly reorients the political commitments of those who affiliate with Global Health in imperial cores, and thus offers important considerations in the wake of Trump Administration attacks on the field.

**Data availability statement:** Per the IRB requirements from the Johns Hopkins University, DK's previous institution, the qualitative data cannot be made publicly available. As some interviewees criticize their institution, funders, and others in the field who potentially can influence their employment, this data cannot be released to jeopardize their employment status with their university and among their colleagues. Given that the deidentified data still has the potential for identification of interviewees based on project descriptions and self-positioning themselves at this institution, the IRB has requested this data not be made available. Nevertheless, data can be requested from the Johns Hopkins Bloomberg School of Public Health IRB at BSPH.irboffice@jhu.edu.

**Funding:** The authors received no specific funding for this work.

**Competing interests:** The authors have declared that no competing interests exist.

## Introduction

Across the growing literature on "decolonizing Global Health," a central predicament has emerged: whose responsibility is it to "decolonize" what, and how should different actors go about different actions considered to be "decolonial?" Seeking to aid Global researchers and practitioners in becoming "allies" [1,2], the dominant discourse underlining common thinking on the global North's role in "decolonization" can be summarized by Wispelwey et al.'s assertion: "a decolonial global health landscape cannot materialise from those whose social positions are settler, White, or wielders of power and influence within untransformed global North institutions" [3]. In other words, through proliferating dissemination of decolonial thinking, responsibility for conceptualizing, leading, and undertaking change is on actors from and in the global South with counterparts in the global North seen as accomplices—an inverse of traditional relationships in the field.

Given the history of Global Health, this is a logical and fruitful departure point for change. In a field grounded in coloniality [4] and productive of vast epistemic injustice [5], those who have been cast to the margins of the field, are closest to manifestations of health inequity, and have the most intimate relation with other epistemologies and ways of being should be privileged actors in efforts to change unequal political dynamics. However, these earnest visions of responsibility are ones that depart from particular ideologies of power. Critiques of ideological power (who shapes thinking and whose knowledge is taken up) and symbolic power (tacit modes of Western domination reproduced through habits of doing Global Health work) disproportionately shape how this responsibility is conceptualized. Conceptualizations of change dominantly depart from these bases, creating normative theorizations of decolonization that center identity and epistemology. That is, it is widely believed the field can be changed through shifting thinking, desires, and actions that then lead to structural changes.

As anthropologists and other critical scholars have revealed, flows of money have, over the past two decades, shaped how ideological and symbolic power manifests in Global Health [6–8]. While funding has begun to shift as donors increasingly give directly to global South institutions in response to historic calls for change, recent thinking about funding as highlighted in the decolonizing Global Health literature identifies how much further these shifts need to go [1,9,10]. This recent thinking elucidates the vast gaps in knowledge about what actions would be needed to change the *material* structuring of the field, including whether giving from Northern sources directly to the South creates enough change [1,11,12]. In other words, it raises different questions as to how global North institutions still largely dominate the field *through* money, how they have compelling financial interests to resist current calls for change, and how flows of funding influence how change is being conceptualized and what it can do.

In this article we argue that conceptualizations of efforts to change Global Health can benefit from examining resource structures and material flows through how those who operate within them make sense of and relate to the contradictions in their work. We do this through describing ideology and financial structures at a major school

of public health in the United States—a major center of Global Health ideological and financial power—through "studying up." As Laura Nader defines, studying up is an anthropological approach that investigates the cultures and financial structures of those with power, with the goal of illuminating how these structures are shaped, and better understanding their makeup and influence [13]. While we originally set out to examine the different discourses of decolonization among these faculty, reflections by these faculty on how internal and external funding structures impact efforts and theorizations of change led to a different line of questioning. What is missed when we think about ideological and symbolic power as separate from material flows of resources? How does connecting these two forms of power alter how we think about the different roles that different kinds of people can have in creating decolonial change?

Through outlining material flows in a powerful Global Health academic institution in the United states, faculty experiences with these structures, and the affects this structuring creates, this article adds to both the analysis of material and symbolic power for the contemporary movement to change Global Health [1,9,14,15]. By further elucidating a material analysis of power and more clearly connecting how it shapes, complicates, and extends the ideological and symbolic critiques already well developed in the field, we offer new perspectives on the question of what it means for global North workers and institutions, to "decolonize Global Health" [16]. At the center of this story is the "soft money" funding structure of academic public health dominant in the United States—a model of institutional financing that requires faculty to procure their own grants from funding agencies to pay for their projects and their salaries. Despite widespread condemnation of and frustration with this structure by faculty in our sample, and how prevalently this system powerfully shapes how faculty do Global Health work and conceptualize change, it is only just being located as a key site for attention [17].

Our examination of "soft money" structuring in this institution elucidates how colonial relations in the field are configured and reproduced through a financial system that dictates what actors are able to do and think about change. Through this materialist assertion, we contend that changing the field will require going beyond only listening to and amplifying the voices of the South. Rather, in centering the complexities the funding structures exemplified by the one elaborated in this study create, we illuminate a profound contradiction: while some global North individuals and institutions are well placed to able to demand and create changes that could significantly reshape the field—and, at least rhetorically, see these changes as logical and desirable—these changes fundamentally go against institutional and personal financial interests. Considering this structuring and the subjugating conditions it creates, we show, demands a rethinking of what change is, who is responsible for action, and more. In the remainder of this article, we outline the experiences of "soft money" in a powerful academic institution in the United States in order to illuminate and contend with the materiality of power engendered by this kind of structuring, the distribution responsibility for change, and this contradiction at the heart of the decolonization movement.

## Methods

In 2023, we conducted semi-structured interviews with 30 faculty members who work in Global Health research at a major school of public health in the United States (Table 1). Recruitment of participants was done between February 15, 2023, and July 1, 2023. All participants were explained their rights and protections before the interview and gave oral consent. To protect the study respondents in the event of a data leak, oral consent was required by the IRB so that the name of interlocutors could not be identified. Further, the name of the granting IRB will not be shared to protect the identity of the school this study was conducted at. All interviews were conducted via Zoom or in-person and recorded via Zoom or on personal devices. Data analysis was done in Deedose coding software following the tradition of grounded theory [18]. DK conducted all interviews with occasional help from AB. DK and AB read and coded all interviews independently.

The qualitative data collection was ethnographic interviews following the outline of JP Spradley [19]. In these semi-structured interviews, participants were prompted to explore their journey to this elite institution, reflect on the impact of their work, and define what decolonization means. While each interview was unique and iteratively guided by the interests of the participants in these different topics, the core theme of decolonization was covered extensively. Responding

**Table 1. Demographic characteristics of interviewees.**

| Sample Characteristics[1] | Demographic Distribution (n = 30) |
|---|---|
| **Sex** | |
| Female | 18 (60%) |
| Male | 12 (40%) |
| **Academic Rank** | |
| Research Associate[*] | 5 (17%) |
| Research Scientist[*] | 9 (30%) |
| Tenure-Track Professor | 7 (23%) |
| Full Professor | 8 (27%) |
| Professor Emeritus/a | 1 (3%) |
| **Race/Ethnicity** | |
| White | 19 (63%) |
| Asian | 5 (17%) |
| Latin | 2 (7%) |
| Middle Eastern | 1 (3%) |
| Indigenous | 4 (13%) |
| Black | 2 (7%) |
| **Nationality** | |
| American | 21 (70%) |
| International | 9 (30%) |

[1]Variable Distributions are reported as n (%) unless otherwise specified.

[*]Non-tenured positions.

to the flow of the conversation, in different ways and at different points of the interview we asked questions surrounding decolonization and its related discourses. For example, we asked when did you first learn about "decolonization"? What were your first feelings when you heard about "decolonization" in Global Health? Do you keep up with the decolonization literature? If so, how much, if not, why? Can you define "decolonial Global Health"? Do you see decolonialism in your work or aspire for it? Questions such as these prompted extensive conversations on the word, its meaning, and change in Global Health. This questioning on decolonization was consistently linked to the soft money structure of this institution and Global Health funding flows more broadly by participants—which is the inspiration for this article. As such, iterative follow up questions about their views on the institution's funding, its effects on their work, and how it hindered change were asked. A complete but camouflaged interview guide can be reviewed in S1 Text.

In this politically charged project that contends with institutional power and geopolitical relations, the positionalities each of us departs from that influenced how we collected, analyzed, and now are sharing this data must be acknowledged. DK is a former public health student at a soft-money institution, but now is in a hard money position—or, is fully funded by their institution—at another elite global North institution. AB is a researcher from the global South whose previous work has been in Non-Governmental Organizations based in the global South and is currently a student at a global North institution. Our experiences with and current ties to this kind of financial structuring of some global North Global Health institutions impacts how we see and interact with this data, and it must be noted that our analysis here is not the only read of the data that could have been created.

## Results

Across interviews, faculty dominantly saw "decolonization," loosely defined, as the responsibility of their global South partners where their work was conducted. Time and time again, interviewees would tell us their role was to "center," "lift up,"

or, most of all, "follow" their global South colleagues on a path to "decolonization" while what "decolonization" entailed was left undetermined. At the same time, all interviewees linked decolonization, or the lack of change overall, to Global Health funding structures and believed that change could not be made until these systems shifted. Although "decolonization" was a point of contention, confusion, and discomfort in the interviews, the problems with funding in the field broadly and their work specifically was something faculty readily and enthusiastically outlined. However, most also saw themselves as powerless to make these changes happen given the size of the structures, their historical embedding, and the fact that the same faculty were materially dependent on them for their livelihoods. A curious pattern thus emerged: while faculty felt as though control and power of resources and projects should be shifted to the South through "following the South'' or "listening to the partners," their own role in this project was dominantly left undefined as they perceived the systems they work in as too large and massive for them to change.

As will later be argued, this central finding of our project maps onto and reflective of the paradox alluded to in the introduction of this article, as well as the lack of attention to materiality in conversations of "decolonization." To establish the foundations of this argument and further outline our data, we organize this section in four parts. First, the soft money structure of the institution is outlined through the words of faculty themselves. We then turn towards the effects of this structure on different faculty members and the widespread complaints against it. Finally, we highlight how these faculty themselves saw this system as an inhibitor of changing their institution and Global Health more broadly, as well as the anxiety the realities of "decolonial" change are producing.

## Mapping soft money

In a soft money institutional funding structure, the path of money begins beyond the university. A small number of faculty at the institution we studied had their full salaries covered by endowment gifts from major wealthy individual donors. Like most other schools of public health in the United States, tenure-track faculty at this institution had a small amount of their salary (around 20 percent) paid for by the school for their work teaching and on committees. The rest they had to raise from grants. Non tenure track faculty generally had to cover their entire salaries from grants. Most faculty were on multiple projects that together added up to a full salary. Grant funding flowing through the university came from a variety of powerful actors. Major funders included the United States National Institutes of Health (NIH), the United States Agency for International Development (USAID), the World Health Organization (WHO), and the Bill and Melinda Gates Foundation. In order to find their salaries, faculty have to win grants or be a part of projects that have won a grant. In each of these grants, the institution gets a preset percentage of money on top of that grant for the institution called indirect costs. While some grants had indirect costs as low as 10 or 15%, this university had negotiated an indirect cost rate of 63%. One senior faculty member explained how this works:

*So, the basis of this is: [the university] says, if we're going to manage Daniel's $100 grant, we want the NIH to give us another $63 to turn on the lights and support all the other stuff that has to be done. So, it's not just about Daniel's $100 he's going to spend on the program or research. [The university] needs another $63. It's a huge tax. A lot of universities do this and it's how they build the buildings and turn the lights on.*

While this American university received 63% of NIH grants, there were different levels of indirect costs allocated to universities and partners in the global South that were engaged on those same grants through subcontracts. Reflecting other explorations of indirect costs (IDCs) [6,10], The same professor explained:

*There's an NIH regulation that says that the IDC [indirect cost] rate for any foreign institutions must be capped at 8%. So that when I do subcontracts to partners overseas, I can't transfer that $63. I can only send them $8 on that whatever $100 I get. These are institutions that don't have deep pockets already. These are institutions that don't have the lights*

*turned on or need even more infrastructure to build up. It's a really limiting issue, and it's unconscionable to me to treat these foreign institutions like this. They have to figure out how to turn the lights on some other way.*

The remainder of the grant money not taken out is then used to run the project, support international partners, and pay the salaries of any faculty member who works on the project. Most faculty are on multiple projects that together add up to a full salary. Tenured professors, tenure track associates and assistant professors, or other faculty who teach also have the opportunity to get up to 20% of their salary paid for by the school for teaching classes to graduate students.

**The effects of soft money**

As one tenured professor described it, this system created a dynamic in which scholars were "hunters with a license:"

*You're this person who goes out and hunts for grants. If you get them, you survive. And if you don't, you don't. They're happy for us to go out. But if you don't get it, you don't stay here. That sounds cruel, but it kind of, I think, actually produces a sort of intellectual edginess to our ideas. Were very competitive for ideas and money and because we don't survive if we aren't.*

While this particular professor stated one of the benefits of the environment, many more stated that the soft-money grant hunting ecosystem limited and dictated their work. This view was particularly prevalent amongst non-tenure track faculty. One master's level non-tenure track faculty member described the structure of funding for their position:

*We are not paid by the School of Public Health at all…As individuals we have to look for new projects to live. If we are funded only from one project and something happened to this project and it suddenly stopped, I will not get paid for the next month. So, we always have to have multiple projects to assure that we have some source of income all the time. Since we are already multitasking and are distracted, we cannot fully commit to a project. I think the combination of those really distracts us from focusing on making our lives better and impacting the countries we work with.*

Senior level faculty who often serve as PIs on projects to fund themselves, their assistants, and partners abroad readily recognized and critiqued this phenomenon as an unfortunate aspect of their jobs. Many stated that they spent more of their time managing grants than doing their research. Others told us that constantly having to find grants was mentally and emotionally exhausting, or contributed to burnout.

The most prevalent of this group's critiques and anxieties, though, centered around how this system impacted others they were either responsible for or had a relationship with–particularly international partners. One tenure-track faculty commented:

*Our donors say, "we're interested in investing in X country, and we're not interested in country Y." And that dictates who we work with. And so, you see that relationship going from us to partners and that project is not an original intention of the partners. But if we want the funds, we pretty much have to comply with donor priorities.*

Others told stories about how they had to close projects or let long-term partners down because they could not secure funding or donors changed their interests—which in turn led to direct material consequences for their partners and the communities they work in. The same professor gave an example and pondered the consequences of it:

*I've been working around family planning in this nation for years. One of the major donors pulled out and decides it's not no longer a priority country. And so, then, this relationship and this collaboration had no funds to survive while both parties, the partners and [the school of public health], see the value of continuing those relationships. It takes trust and*

*time together to know how to work together. That harmed, I think, the capacity to build more equal relationships with them.*

Junior faculty, especially those working in satellite sites or in the field, also commented on these phenomena. As master's level non-tenure-track faculty bluntly put it, *"once I started working abroad one of the first things that hit me was the funding cycle for the projects that we work on are ridiculous and stupid and seem designed to fail."* One working in a low-income community recalled a particular time they experienced the consequences of this "cycle" directly:

*I was doing a study for COVID-19 where we were going out into the community and testing people for antibodies. At one point the funding just dried up. We just couldn't get the antibody test anymore so we had to quickly buy new ones. It was too expensive and then we went over budget. Then we can't support some staff and had to let some people go because funding ended for certain projects. People are only to be paid as long as we have more studies that can be funded.*

Tasked with the day-to-day operations and data collection for these projects both this respondent and others stated that this system creates a hierarchy in which they most intimately see and must directly navigate the consequences of this system but they can do little about it. As another master's level faculty explained:

*These PIs have certain deadlines they have to meet. They always have to be applying for more funding. And I think maybe they get into this mindset and what the project does matters less. I guess I can maybe say "well, I don't think this project is very helpful." But I'm still at the bottom of the totem pole.*

Another junior faculty member witnessed the flip side of this donor-driven model. A part of a team tasked by the Gates Foundation to lay the foundation for the introduction of a vaccine into a nation struggling with socioeconomic and political turmoil, this researcher had to "badger" top officials in the nation to gain information despite questioning "why would we introduce this new vaccine that's going to double the price of [this nation's] whole vaccine program?"

*This research is not useful for them. We're like calling them up or texting them on WhatsApp and we don't even have anybody in the country. It feels so uncomfortable like just trying to cold email people in some cases. You know, we're [redacted university], like, hopefully they open the email. And that's a bad feeling that we're writing on that premise to these people who are very senior. We have to take an hour of their limited time to talk to us about a health issue that's not a priority for them. So, it's been very hard getting these interviews, and then there's this feeling of like "you guys are…why are you coming from outside and asking us these questions, like don't you think we know how to make decisions for ourselves. Don't you think we have good decision making here?" And they do. The last call that put me over the edge was with somebody who's very well-known, and he basically called this whole project imperialist. I think that has a lot of merit and I felt so ashamed.*

Some faculty noted how the mindsets produced by this ecosystem in which "everything we do, every minute of time we spend on something at this institution has to be grant funded" are now beginning to spread beyond this institution through the networks across the world the school has fostered. A senior-level professor observed this in advising doctoral students studying at universities beyond the United States:

*I'm increasingly seeing some of the unfortunate aspects, the biggest one of which is that people are pursuing research agendas based on what money is out there and not based on what they think are relevant and interesting issues to investigate. It makes me sad that they [graduate students] don't get to study what they want because they have to*

*chase money. I'd rather facilitate them to figure out "here's what I care about in the world, here's what I find intriguing," and then they can figure out a way to do it and get the funds. It's hard in a soft money environment where they think they have to go find the funds rather than figure out what they want to do.*

### Hindering social change

As many dislike this ecosystem where their work is "fulfilling the goals of the grant that feels like a waste of time," as one research scientist put it, this is something they readily want to change. However, as numerous faculty explained, not only does the soft money structure make it difficult to resist the soft-money system itself, but also to resist much broader change in the field. The elite university studied here is, like the vast majority of US universities in the 21st century [20], a business. It is a private university that needs a financial plan, spending limits, and rules to sustain its monetary growth over time. Not only does this create the "hunter with a license" dynamics seen above, as the institution needs grant money and the overhead they take from it to survive, but also a social system where protection of that money and the mechanisms to get it are most important. A non-tenure track but senior member of the department summarized value in this system:

*It's a fiscal pressure. For faculty members to be productive, you have to bring in money. I would prefer that faculty members are fulfilled, they're doing good work, and that the metric of their productivity is their fulfillment as a benefit to the world. But that's not what's measured. What's measured is how much departments and how much faculty members bring in revenue and to me that's been the most disheartening thing.*

The same faculty member later commented on a paradox that rests at the heart of the position of high-income country academic institutions powered by soft money. While the school of public health *"would love for you to train and work with a partner and in a three to five-year time frame that the partner then takes over the grant and project,"* she states, the school *"needs you to bring in grants."* A number of faculty pondered this core tension between the ideals of global public health centered around sustainability and the soft money funding model that demanded they procure grants.

Respondents largely agreed that change at the school of public health—whether it be initiated internally or by outside pressures—would likely be limited if it disrupted that bottom line of the broader university. *"As long as they get their 63%,"* a professor familiar with the IDC system stated, *"I don't think they would care if their partners got 63%. All [the university] cares about is getting their 63%. What they would care about is going from 63 to 50 percent."* Another senior professor summarized this sentiment bluntly, *"the political economy of this, the soft money environment, means that [the university] will try to protect itself. It may give a nod to these changes, but actually, because of what the money demands, it will resist it."*

Thus, scholars interviewed widely argued that the impacts of soft money hindered any efforts to "decolonize" the field. A senior scholar eloquently summarized this sentiment:

*I don't really feel like the problems of the field are going to be properly addressed with the current funding structures that we have. I mean, a lot of the funding for the work that we do is coming from foreign aid type support groups. People want to see numbers that they can then campaign on and talk about that they've supported. So, I don't really see that being dismantled and rethought anytime soon. But I think that that's what would have to happen. It would have to be completely reconceptualized, rebuilt, restructured in a more functional and productive way.*

However, while most agree that fundamental changes to the funding structure are needed, how faculty members feel about change vary. The contradiction between ideals and business that underpins this institution is intimately experienced by faculty members. As their livelihoods depend on procuring grants, fundamental change to this system that would

benefit their partners could potentially harm their ability to provide for themselves and their families. One faculty member pondered contradiction:

*There's so much focus on releasing papers and getting grants. You have to get grants because if you don't get grants, you have to leave. You have to live. You have to feed your kids. You have to pay rent. It's not that there's anything wrong with the fact we have to do this. I feel conflicted about it sometimes because it's all about just getting the grant. I want to do other things. I want to do quality research.*

### Anxiety about funding shifts to the global south

This concern with their personal livelihoods begins to unveil the profound anxiety felt by faculty about the future of the field and the potential impacts of decolonization on them. When conversing about the paradox of their personal interests versus ideals of equity and sustainability, faculty expressed fear about a world where they no longer win grants, their institution gets their needed indirect cost money, and they potentially get their jobs cut when funding shifts to institutions in the global South that are currently subcontracted as their partners. Most of the time this fear generated statements such as "I don't know" and produced visible discomfort. Other times faculty theorized or pondered a world where funding could shift but it would not threaten their jobs. For example, faculty contemplated how "it would just be a different way of being funded" in which partners in the global South would sub-contract them to grants. Overall, faculty felt that partners around the world would continue to work with them even if these partners were fully funded on their own. One faculty summarized this common idea:

*Our partners on projects are keen to do different kinds of work than they have historically been expected to do as part of the research agreements that they participate in. However, that doesn't necessarily mean that they desire to do every aspect of the work independently and on their own.*

Seemingly to move away from this uncomfortable contradiction, many faculty point to funders or university administrators as the true place where blame should be directed. A common sentiment across the participants was that they as faculty were powerless to do anything to change the soft-money structure at their institution or the wider Global Health funding ecosystem. Many faculty argued that these structures are too big and embedded to even begin shifting, that it is the responsibility of people at "higher levels," or that they do not know how those shifts would happen. These points were summarized one respondent:

*I don't think the researchers have the power to change much, because the reality is an institution like [this university] is soft money. We respond in many ways to our donors' areas of priority. They set the priority, whether it's for us or the global South. But in general, those lines of priorities are from donors from the global North. And I don't think at [this university] we really have the power to change that.*

While these sentiments can be used to deflect responsibility for change, as will soon be argued, they have merit. When asked how change could be created in the field, perhaps the most high-profile member of the school we interviewed stated it can only be made by particular people in particular ways:

*It's through personal contacts. And I know someone that takes me to who I want to talk to about what I need. To talk to [a major donor], you have to go through the president. He has his council of deans, so you move it up to the Deans and they see it. I have tried doing it. I've just met with the [deans].... The question is, who's in the power structures that can take it to the next layer? It's me, I need to be that point person and I'm not afraid to do that. So, I am setting up meetings. I'm talking to them more. But it's the deans who oversee the fundraisers, the philanthropy people, and we need to*

*get them more engaged. Somehow, we have to keep bringing up our priorities. If you're a dean, you're looking at the financial well-being of your school and your students. You're a little bit less attentive to what [the institution] is doing in Uganda or Zimbabwe. You know about it, you have your advisory report on those places, so gets some attention. But you have to work on getting their attention constantly.*

## Discussion

Global Health, in addition to an ideological frame [21,22] and an action field [23,24], is a superstructure intrinsically linked to neoliberal global capitalist worldmaking [25,26] that dictates how and where large amounts of money flow [27]. By outlining how Global Health at this powerful academic institution is a business where resources flow in particular ways in which more benefits, protections, and influence are granted to those most adapted to "playing the game", we have sought to open new conversations about social change in Global Health and who is responsible for it. Importantly, here we must note that our research does not overview all types of funding in Global Health academic institutions. Rather, like the vast majority of schools of public health in the United States and to an extent abroad, Global Health *research* funding is what is sought after and contended with by these scholars. In contrast, *programmatic* funding—money given to run specific interventions and public health programs by the likes of PEPFAR and the Global Fund—has different monetary flows. While some scholars at the highlighted school of public health do receive funding for programmatic-specific work, this article explores the flows and consequences of research funding for Global Health common to elite American universities.

In this discussion section, we highlight how detailing the material reality within one of the most elite global North academic institutions allows us to see one of the many ways in which power in the field is materially constructed and reproduced; one that creates a paradox of decolonial change for the actors in this and similar institutions and the necessity for reconceptualizing responsibility for change as *ruinous solidarity*.

### The materiality of power

As Tuhebwe et al. make clear, "when funding and decision-making power are centered in the global North at the outset, the stage is set for power imbalances that persist throughout the project life cycle" [28]. As Global Health does not have its own resources and must rely on global North states, philanthropies, and finance capital for funding, money is already politically determined before it reaches PIs at schools of public health. The NIH, USAID, and CDC and their counterparts in other global North countries set the priorities of what research topics, problems, and types of projects will be funded as well as which applicants will win those grants. Philanthropic organizations and foundations are created in the image of mostly white male billionaires or Fortune 500 companies, and largely operate at the whims of the influence and interests of their leaders [29–31]. Both of these original sources for capital, as well as private finance [8], are premised on fixing problems while sustaining the broader political economic system that gives them legitimacy, political power, and economic gluttony [32,33].

Here, we show how actors at this institution are forced to reproduce colonial inequalities of their work when this already politically charged money is acquired in "soft money" funding schemes. As shown, the survival of the individual researcher and the university are intertwined in the "soft money" business model. If grants are not won, the 63% on top of these grants through IDC taxes is not taken by the school of public health, and the bottom line of the university is hurt. Failure to secure grants, then, is not an option for the researcher. Because of this, faculty spend the bulk of their time "hunting" for grants in a constant cycle of pursuing funding, conducting projects, and publishing papers mostly for the aim of securing the next grant and, thereby, the personal income for themselves and their junior colleagues, dynamics others have also elucidated [6,10,28].

Many faculty are thus left tired, burnt out, and discouraged that impacting communities elsewhere is an afterthought rather than the focus of this system. The structure of how they secure their livelihoods not only hinders the ability for them

to form long, multi-year or decade partnerships with colleagues in formerly colonized regions, but also leaves them and their colleagues exposed to sudden changes in donor priorities. As was seen in one faculty member's description of the rug being pulled out from underneath her and her partners in a nation she was working in for a long time, to the funders and school, what mattered in the end was not these relationships, the longevity, or the impact of the work, but rather, first, what the funder wanted to give money to, and, second, that the researcher switched locations and abandoned their partners to continue bringing in money for the school.

This arrangement negatively affects faculty members at these elite schools, disproportionately distributed so that it is worse down the hierarchy of research positions. It also engenders and empowers certain forms of change while discarding others. As some faculty at this institution outline, using the latest buzzword or concept bolster their chances of winning grants and funding required by their institution. Thus, this imperative of being at the cutting edge of the field, coupled with the anxiety born from the contradiction that this arrangement sharpens, incentivizes cosmetic alterations that leave the situation largely unchanged. As they attempt to fit concepts such as "decolonization" into the context of one's research, words from the apparatuses and movements beyond Global Health are reconfigured to signify forms of actions they did not originally refer to. While "decolonization" is thus captured from these other spaces and depoliticized of words [34], the prestige and resources of the researcher and their institution are reproduced. This, as some faculty interviewed showed, was a dynamic they were cognizant of but could not stop given the necessity of winning grants.

In sum, the structuring of this institution based on soft money is designed in a way that maintains inequities. While this is one, albeit centripetal, example of an institutional financial system, the soft money system outlined here is a baseline structure that has been replicated across the world In this financial structure as elucidated through this case, gaining vast grant money and prestige, the financial integrity of elite Global North institutions depends on this system in which their researchers must secure a disproportionate amount of the few grants that are available at the expense of their Global South partners.

## The double bind of change in the global north

In this environment, interlocutors in these soft money positions find themselves in a double bind. Although many are at least partially discontent with this system, understand how it inhibits their work and their partners, and see how it is contradictory to their ideals, they are materially dependent on participation in this system to advance in their careers and create their livelihoods. While many of the interviewees desire change currently being theorized as "decolonization" in Global Health and more equity for their partners, they also saw what changes to the funding system could potentially mean for them. Whether "decolonization" is understood as more ownership over funding by global South actors [1,12], the end of the need for Global Health as infrastructures are developed and complete autonomy over health-related research and interventions is gained in formerly colonized regions [35,36], or the abolition of the global capitalist system that Global Health is supportive of and supported by [26], Northern researchers like our interviewees and their institutions would be affected. If a vast amount of funding is redirected away from global North institutions to institutions in the global South, it could, as interviewees imagine, lead to financial struggles for elite institutions that rely on these grants, a decrease in prestige, and job loss. To thus argue for "decolonizing Global Health" from within this institution and similarly structured one is to advocate against the business model of one's employer and for the redistribution of the money that sustains one's projects, department, and livelihood.

However, the effects of this double bind and the impasse it generates are much deeper. Just as how the structuring of soft money encourages elite capture of the rhetoric of radical programs of change to fit within the established actions and professional habits of these scholars, so too do their actions in response to this dilemma in the end seek to preserve the established structuring of Global Health. Rather than grappling with the positive possibilities of their job ending, as only one interviewee did, faculty theorized the different ways they and their school could simultaneously maintain its size, prestige, and power while giving more resources and agency to Southern partners. In this, faculty theorized a future where

they remain "marketable" and "necessary" to the world where they are "subcontracted" to global South partners. Further, reformist programs focused on diversity, equity, and inclusion, revising syllabuses, or "listening to partners more" become the way faculty—who largely assume they cannot create significant change given the field's size and power—are able to "participate" in the movement that aligns with their ideals without directly threatening their or their employers' livelihoods. In the end, both of these responses sought to preserve the status quo while acknowledging the need for change.

Thus, there is a disconnect between the stated values of these many faculty and the ways that faculty articulated how they could not achieve this in practice because of the structures they were embedded within. While nearly all agreed that power and funding streams should shift to the global South to some degree, none saw themselves as being able to take tangible steps toward this given their embedding within the system and dependence on it, nor saw the powers of the field significantly changing the system due to its entrenchment and self-reproducing flows of money. The best change, as they articulated and tacitly communicated in their assumptions about what was and was not viable, were these small ones they could make within their projects and relations with global South colleagues. In other words, these faculty who benefit *most* from the Global Health financial and prestige system felt *least* able to and responsible for creating material change.

## Rethinking responsibility through ruinous solidarity

In all, what we have sought to demarcate in this section are the material contingencies of change created by soft money financial structuring exemplified by the university of this study. What we find is a self-reproducing system that delineates particular feelings, imaginations, and political impetuses among Global Health researchers subjected to this system. The question thus becomes how to break from this inert, hegemonic, ouroboric political imaginary that displaces responsibility for change onto subaltern experts and begins to build movement that can seriously shift these structures. We argue here this begins with a necessary rethinking of solidarity among Northern Global Health elites—moving from its normative usage as exemplified by our interlocutors into a new frontier of *ruinous solidarity*. Gary Wilder gives a conception of solidarity that contrasts with ideas of solidarity and sets the basis for charting ways out of this sticky situation. This is a solidarity that starts from entanglements, is a practice not a sentiment, and therefore requires risk [37]. Rather than feeling *for* global South actors who should be "listened to" or "uplifted," or exclusively being allies in their demands, this solidarity is a practice born out of an understanding of the ways in which the situations of all workers in Global Health are entangled—in this case inside a global capitalist system and the superstructure of Global Health that functions inside of it. The same material structures constrain both parties and limit their courses of action to those set by powerful and moneyed donors, and thus action on part of agents who want to create change must be pragmatic and contextual while connected in a unified political program.

We combine this conceptualization of solidarity with Jenna Hanchey's idea of "productive ruination" in international development and Global Health [38]. Asking "what might it look like to figure some processes of ruination as *productive* of possibilities for decolonization and justice?" [38]. Hanchey pushes for consideration how potential for otherwise political action, institution building, and worldmaking is found when the desire to reinforce unequal systems is abandoned and recognition that some may lose comfort or opportunity in this change is internalized.

Similar to what Amilcar Cabral calls "class betrayal" [39], *ruinous solidarity* is thus thinking and action on the part of more powerful, Northern actors that embraces the possibilities that emerge in downsizing and, thereby, seeks to aid simultaneously disintegrative and generative falling apart of the field as it exists. Returning to Wilder, the sort of practice that arises out of this recognition requires risk or "renouncing safety and sharing risk, putting oneself on the line by propelling oneself over the line that is supposed to mark an outside" [37]. Taking seriously the distribution of material power in Global Health, the depths of this funding structure, and how global North institutions are fundamentally built to reproduce established power and will neither simply give away that power nor voluntarily risk financial stability, a complete shift in ideology and praxis is required. Current reform-based efforts in many schools of public health are essentially tactics to participate in a movement without reckoning with the fact that "decolonization" would most likely result in loss of jobs, power, and

prestige for global North faculty. Ruinous solidarity moves global North workers and powerholders from the sideline of social change as only allies of Southern counterparts to active co-conspirators for the transformation of their own and other's subjecting conditions of Global Health work.

Reformulating responsibility for creating change as pragmatic, falling on Global Health academics of any positionality in the North as it does their colleagues in the South, ruinous solidarity seeks to open a fruitful realm of political imagination in Global Health to inspire action against the material structures of the field. Rather than displacing responsibility to Southern counterparts, ruinous solidarity acknowledges that marginalized actors must be privileged recipients of change but not solely responsible for changing material systems they are on the periphery of. In embracing the fact some Northern researchers and institutions could lose substantial institutional and personal resources, and this potential aligns with the imagined future of an equitable field that is so desired, the possibilities for material action among Northern technocrats becomes less "radical" and more essential. That is, inertia towards changing the soft money system that limits their work *and* engenders the systemic inequities for Southern counterparts is built when there is acceptance that material changes to Global Health structuring has equal propensity to cause austerity and loss in the North as it does to result in more just, generative, and affirming arrangements for everyone. With the need create material change to the systems that create the symbolic and epistemic injustices in the field [5,9] realized, tangible social movements to dissipate Southern dependence on Northern funders and researchers through reparative funding can be realized. What ruinous solidarity does is provide a political imaginary to break apart the idea that nothing can and should be done materially by Northern elites and, rather, place much of the responsibility of change on them to break apart the norms and systemized material flows that enable them.

## Conclusion

While reimagining Global Health funding has been a central part of decolonizing Global Health conversations, global North experts have been spared from grappling with how the material structuring of the field affects visions of change and how solidarity demands more than symbolic and epistemological commitments. We have shown how soft money funding structures can maintain the inequalities of Global Health and how it is beyond the financial interests of elite global North institutions and actors under this funding model to advocate for real structural change. We have argued that conceptualizations of "decolonization" as cosmetic inclusion or a win-win situation in which global North institutions can maintain prestige and financial resources amidst widespread rhetorical change, and how reconfiguring power in the field will require those who are more advantageously structurally positioned to betray their immediate interests and push demands for change that may affect them and their institutions substantially.

In this, we hope those concerned with reconfiguring Global Health financial structures consider the social grounding, cultural self-reinforcement of the system in which elite institutions will protect their bottom lines no matter the ethical stakes. For global North technocrats, we hope reframing how this system and social change is thought—moving impetus away from deflecting responsibility to others and toward embracing their role in changing their institutions as privileged actors whose loss might open opportunity for gain among the regions they work to improve.

But while ruinous solidarity offers a way to think about changing Global Health, responsibility for it, and what it requires departing from a materialist perspective, it does not provide all the answers. How is this all practically enacted? How can the logic of change among global North actors be shifted to a place where they are ready to challenge the roots of their privilege and livelihoods? How can this partially destructive change be enacted in ways that create the least amount of personal suffering for those impacted by new flows of money? These are questions that must be answered. Further, this study is limited to a single American university exemplary of a typical soft-money scheme. How funding streams impact political imagination and commitments to different forms of change other institutions—such as non-grant funded programs or European structures with a different configuration—in America and beyond must be studied.

As the second Trump Administration's gutting of USAID, shock changes to the NIH IDC rates discussed in this article, and banning of all equity language results in massive culling of Global Health positions around the country, including at this university, the implications of this study and the need for this kind of solidarity are amplified. The contradictions that we elaborate in this paper—the want to preserve the current state of a degrading Global Health research infrastructure but at the same time desiring change that favors Southern colleagues—are likely to become more sharply felt by all who rely on funding from the agencies that are being attacked. The power of rhetorical and other surface level changes such as the adoption of "decolonization" and other buzzwords and cosmetic forms of diversity, equity, and inclusion will likely become unfavorable, and therefore unavailable as a soothing balm for the anxieties the interviewees and others in similar positions. Thus, a shift of focus may be on the horizon: one from more equity and justice focused work to work that is more aligned with the unimpeded continuation of business-as-usual to preserve what is left of US-based Global Health.

In this chaotic moment, ruinous solidarity not only becomes even more imperative as a posture towards changes in the field of Global Health but also offers a needed "both/and" perspective to these changes. Amid massive cuts to the Global Health research infrastructure in which morally-inclined, well-intentioned professionals are losing their jobs, thinking with ruinous solidarity offers a stance where it is the responsibility of US-based Global Health actors to *both* combat fascist attacks on health both at home and abroad *and* to continue working towards change that redistributes their resources and power. In other words, ruinous solidarity demands a fight back against these cuts is necessary to regain power over this process of changing Global Health *while* recognizing that this fight should be for a transfer of wealth, responsibility, and more to full Southern ownership. Avoiding backsliding into preserving an idealized status-quo will require collective imagining of how this moment can be pragmatically *used* to further material changes that have been called for and how resistance to Trumpian fascism can advance the field's generative end [35,36].

## Supporting information

**S1 Text. Interview Guide.**
(DOCX)

## Acknowledgments

The authors would like to thank Svea Closser for her guidance, edits, and commentary from start to finish of this project. We are also grateful to Seye Abimbola for suggestions and edits on both early and later drafts of this paper.

## Author contributions

**Conceptualization:** Daniel W. Krugman.

**Data curation:** Daniel W. Krugman.

**Formal analysis:** Daniel W. Krugman, Alice Bayingana.

**Investigation:** Daniel W. Krugman, Alice Bayingana.

**Methodology:** Daniel W. Krugman.

**Project administration:** Daniel W. Krugman.

**Writing – original draft:** Daniel W. Krugman, Alice Bayingana.

**Writing – review & editing:** Daniel W. Krugman, Alice Bayingana.

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
