## [Decision Letter · Decision Letter 0]

20 Nov 2024

PGPH-D-24-01739

Soft Money, Hard Power: Mapping the Material Contingencies of Change in Global Health Academic Structures

Dear authors 

Thank you for submitting your manuscript to PLOS Global Public Health. After careful consideration, we feel that it has merit but does not fully meet PLOS Global Public Health’s publication criteria as it currently stands. Therefore, we invite you to submit a revised version of the manuscript that addresses the points raised during the review process.

*Comments from PLOS Editorial Office: We note that one or more reviewers has recommended that you cite specific previously published works. As always, we recommend that you please review and evaluate the requested works to determine whether they are relevant and should be cited. It is not a requirement to cite these works. We appreciate your attention to this request.*

We look forward to receiving your revised manuscript.

Kind regards,

Andreas K Demetriades, MBBChir, MPhil, FRCSEd, FEBNS.

Academic Editor

Journal Requirements:

Additional Editor Comments (if provided):

Reviewers' comments:

Reviewer's Responses to Questions

**Comments to the Author**

1. Does this manuscript meet PLOS Global Public Health’s publication criteria ? Is the manuscript technically sound, and do the data support the conclusions? The manuscript must describe methodologically and ethically rigorous research with conclusions that are appropriately drawn based on the data presented.

Reviewer #1: Yes

Reviewer #2: Partly

Reviewer #3: Yes

2. Has the statistical analysis been performed appropriately and rigorously?

Reviewer #1: N/A

Reviewer #2: N/A

Reviewer #3: N/A

3. Have the authors made all data underlying the findings in their manuscript fully available (please refer to the Data Availability Statement at the start of the manuscript PDF file)?

Reviewer #1: Yes

Reviewer #2: Yes

Reviewer #3: Yes

4. Is the manuscript presented in an intelligible fashion and written in standard English?

Reviewer #1: Yes

Reviewer #2: No

Reviewer #3: Yes

5. Review Comments to the Author

Reviewer #1: Thank you for the opportunity to review this important article, where Kurgman and Bayingana present an intriguing paradox - while academics situated in elite Global North institutions agree that funding systems should be fundamentally rearranged to provide direct funding to institutions in the Global South, this shift threatens their livelihoods and they fear the system cannot be changed. This paradox makes us rethink what it really means to “decolonize” Global Health, and what that might entail in practice.

Overall:

1. Make sure you are consistent about capitalising/not capitalising Northern and Southern

2. There are several run-on sentences in this paper - to increase readability for non-expert audiences, I would suggest making some of them shorter. E.g. see page 2 paragraph 3, sentence starting with: “While funding has begun to shift…”

Abstract:

1. I think it’s important to note somewhere in the introduction that this is a case study from a single institution (albeit major institution)

2. Suggested simplification of wording in the last sentence: instead of “As thinking and action on the part of elite, northern actors that embraces the possibilities that emerge in loss of resources and helps facilitate this transfer that materially and symbolically disadvantages them”, perhaps use some wording from your discussion: “The disconnect between the ideology of these faculty—nearly all agreed that power and funding streams should shift to the Global South—and the ways that faculty articulated how they could not achieve this in practice because of the structures they were embedded within,…”

Introduction:

1. Page 2 paragraph 4 - please define “studying up”

Methods:

1. Page 4 paragraph 2 - please define “hard money position”

Results:

1. Page 4 paragraph 3 - please change “powerlessness” to powerless

2. Page 4 paragraph 3 - “the fact that they were materially dependent on them for their livelihoods.” - who is “they” and who is “them”?

3. Page 5 paragraph 3 - by indirects do you mean indirect costs?

4. Page 6 paragraph 1 - remove BSPH (I presume the university isn’t supposed to be named)

5. Page 7 paragraph 5 - is there a reason the quote in this paragraph isn’t separated and italicised like the other quotes are?

6. Page 8 paragraph 4 - remove BSPH

7. Page 9 paragraph 1 - please put the full form of “IH” in IH Department

8. Page 10 paragraph 2 - remove BSPH

9. Page 10 paragraph 4 - right after the sentence where you say “faculty expressed fear about a world where funding dramatically shifted” - I suggest adding a line to express what that shift means. Is it that funding is routed entirely to institutions in Global South, at the expense of limiting IDC for the Global North? Or is it that both parties “get their 63%”?

Discussion:

1. To contextualise this research study in light of recent political events in the US - you can perhaps note somewhere in the discussion that budget cuts to research funding could exacerbate issues that have been highlighted in your study

2. Page 12 paragraph 3 - I’m not sure if you’ve used “GN” for Global North before - please replace with Global North, or explain the abbreviation somewhere above

3. I suggest adding a line to critique the potential limitations of your approach - this is a single centre study with a predominantly white American sample being interviewed, which could skew the results. However, this provides a fantastic starting point - it would be really interesting to see how the themes that emerged from your study compare to those in other institutions.

Conclusion:

1. Page 17 paragraph 1 - please put the full form of DEI

Reviewer #2: Soft Money, Hard Power: Mapping the Material Contingencies of Change in Global Health Academic Structures

Review comments to Authors:

This paper reports the findings of an interesting qualitative study. I appreciate that the authors are directly exploring factors that perpetuate inequities and neocolonialism in global health. Would recommend publication if the authors complete a significant revision, per my comments below:

1. I do not agree with the underlined words in the following statement, “how Global North institutions still largely dominate the field through money, how they have compelling financial interests to resist current calls for change, and how flows of funding influence how change is being conceptualized and what it can do remains unexamined.”

The influence of funding flows on change (or lack thereof) in colonialist power structures is, in fact, an active area of scholarship and critique in global health. Structural issues related to funding flows and incentives that reinforce Global North predominance have been examined by several different authors. I’d encourage these authors to read the papers below (and review the reference lists) and cite some or all of these references in your own paper at the appropriate points:

- Haberer J and Boum Y. Behind-the-Scenes Investment for Equity in Global Health Research. New England Journal of Medicine 2023;388:387-390.

- Oti SO (2024) Towards authentic institutional allyship by global health funders. PLOS Glob Public Health 4(3): e0003024. https://doi.org/10.1371/journal.pgph.0003024

- Cakouros BE, Gum J, Levine DL, Lewis J, Wright AH, Dahn B, Talbert-Slagle K. Exploring equity in global health collaborations: a qualitative study of donor and recipient power dynamics in Liberia. BMJ Glob Health. 2024 Mar 13;9(3):e014399. doi: 10.1136/bmjgh-2023-014399.

2. I note several places where copy-editing would help improve the readability of this manuscript. Below are some examples (note that I did some copy-editing in CAPS, but other changes were not retained when I submitted this review through the PLoS GPH website):

- In the first sentence of the Abstract: “In the proliferating conversations about decolonizing Global Health, a basic assumption has been THAT Global South actors WOULD/WILL BE the primary conceptualizers, leaders, and undertakers of change with Global North counterparts thought of as allies or accomplices.”

- Second sentence of the abstract: “…this article argues that examining resource structures and material flows complicates assumptions about whose responsibility it is to “decolonize” what and how different actors should go about different actions considered to be decolonial.

- From the introduction: “With making power itself an object of study, this anthropological approach that investigates the cultures and financial structures of those in power, with the goal of illuminating how these structures are shaped, enabling better understanding their makeup and influence [11].

- From the results: “…most also saw themselves as powerlessness to make these changes happen given their size, historical embedding…”

I’m not going to mark all the copy-editing that is needed in this manuscript, but I encourage the authors to give the manuscript another thorough revision before resubmitting to improve its readability and clarity. (Might be worth paying a copy-editor to do this for you, perhaps a student who is a strong writer but not an expert in this field, who can help improve clarity for lay readers.)

3. Please make sure you are correctly using the words “Affects” (a verb) and “Effects” (a noun).

4. I’m not convinced that “Global Health” needs to be capitalized throughout this manuscript. To me, this is distracting, but I will accept it if the authors feel strongly.

5. I really like and appreciate these questions as anchoring points for this paper:

“What is missed when we think about ideological and symbolic power as separate from material flows of resources? How does connecting these two forms of power alter how we think about the different roles different kinds of people should have in creating decolonial change?”

6. Is “Marxist analysis” a specific anthropological method or framework? It is not cited, nor is it clear how/why this particular type of analysis is relevant or the best choice for this paper. Fine with me if you omit it in the next version, but if you choose to retain it, please cite/explain why.

7. You refer to “BSPH” as the school or “the IH Department” where the interviews were conducted, but in other parts of this manuscript, you seem to be taking pains not to identify the “major Global North school of public health.” In one of the quotes, the respondent says “we’re Johns Hopkins.” All such identifying information should be removed to protect the anonymity of your respondents.

8. You refer to “the IDC system” without explaining what that is; please do not assume that your readers will understand this niche terminology. Also, please cite the Haberer and Boum piece when referencing inequities in indirect rates!

9. In your Discussion, you state: “By outlining how Global Health is a business where resources flow in particular ways in which more benefits, protections, and influence are granted to those higher up on the hierarchy, we have sought to open new conversations about social change in Global Health, what it must do, and who is responsible for it.” But this is a significant overstatement of what you have reported in this manuscript, and it undercuts the value of your work. You have not described the entire field of Global Health as a business, nor have you described the hierarchy of the entire field. Doing so would take many, many papers doing what you have done here as well as many other types of analyses. What you have done is more narrow but very important: you have provided primary data from Global North actors at a U.S. public health institution that captures their firsthand experience in a soft money funding structure that perpetuates power disparities and structural inequities. You don’t know, from your data, whether this description captures a broader phenomenon, but you can call for more research like your paper to explore that question. I urge you to rewrite the first paragraph of your discussion to summarize your findings without making such a grandiose claim.

10. You repeatedly refer to the Global North institution as “elite” or “most elite,” which I find off-putting. What do you mean by that? And why does it matter? Is the “elite” nature of the institution relevant to your findings? I find myself wanting you simply to report that you conducted this analysis at a public health school in the U.S. , and that, from your own positionality, you suspect that the phenomena you have identified may reflect common issues at many public health schools in the Global North (and you actually specifically mean public health schools in the U.S. – from my own research, I know that at least some Canadian and European institutions handle indirect costs and funding structures differently from U.S. institutions; you want to be careful to provide an appropriately limited description of what you claim to be studying—and to have learned—here).

11. Again in the discussion, you overstate what you have demonstrated in your results, writing: “Here, we show how Global North actors are forced to reproduce colonial inequalities of their work when this already politically charged money is acquired in “soft money” funding schemes. As shown in the data, the survival of the individual researcher and the university are intertwined in the “soft-money” business model.” You did not conduct a study that explores enough Global North institutions (which includes many countries beyond the U.S.) to be able to make this claim. Many Canadian universities offer hard-money positions for their public health faculty. I think the same may be true of European institutions. It may also be the case that some U.S. universities offer hard-money positions for public health faculty; you do not know. Make sure that you describe your findings accurately and appropriately. Overstating your results discredits the validity of the entire study. You need to change this throughout the discussion.

12. Your discussion section is undercited. Several of the dynamics and power differentials you mention have been discussed and called out openly in other publications – important to include those and honor the work of others who are doing this anti-colonial work. Please revise with a more thorough effort to cite the work of others who have called attention to these problems (see the papers I cited above as a good starting point for familiarizing yourselves with this literature).

13. The last paragraph of the section entitled “The Materiality of Power” reads like an opinion piece. It is not very grounded in the results you present earlier in the paper. Your tendency to overstate the breadth and reproducibility of your findings is not doing good service to the important work of this paper. Please revise to focus your discussion more on your own results and on how those results add to our understanding of these issues in global health.

14. The following excerpt from the Discussion feels very important to me, but it is difficult to understand: “…the end of the need for the field as it exists as infrastructures are developed complete autonomy over health-creating research and interventions is gained in formerly colonized regions.” Please reword for clarity.

15. This is an excellent sentence: “To argue for “decolonizing Global Health” in Global North institutions, then, is to advocate against the business model of one’s employer and for the redistribution of the money that sustains one’s projects, department, and livelihood.”

16. This is opinion only, for the authors’ consideration: I think the section entitled “The Double Bind of Change in the Global North” is very well-written and powerful. I’d suggest moving it up in the discussion and perhaps cutting some of the rest of the discussion section to emphasize the points made in this section. It feels strongly grounded in your own primary data and highly relevant to the takeaway points that you intend to make in this manuscript.

17. The section “Rethinking Responsibility through Ruinous Solidarity” is interesting…it is written differently than the rest of the discussion, with many more direct quotes. I think the “ruinous solidarity” concept is worth raising, although I would prefer that you reach it more directly, with less lead-up and fewer direct quotes of other authors. I think, however, that my critique has to do with my training; I am not an anthropologist and prefer more direct writing. So, I give you this critique to consider: I’d like for this section to be pithier, which I think would make the contribution of the “ruinous solidarity” concept easier for your readers to grasp and therefore more powerful. But I defer to the authors on this point.

18. Having reached the end of your paper, I do think this is an important contribution to the literature and worth publication after revising, but your analysis is not about the Global North. It is specific to the U.S. and to a single U.S. university. Every funder you mention is a U.S. funder (NIH, CDC, USAID, Gates), and the issues you flag are unique to U.S. soft-money institutions and the way that those U.S. institutions manage/negotiate indirect cost coverage with their cognizant federal funding agency. The dynamics that your respondents describe are not universal to wealthy Global North institutions, and you need to be careful not to say so. Please revise the paper to indicate that your analysis focuses specifically on a U.S. university.

Reviewer #3: Dear Authors,

This is a bold and a very exciting manuscript. I am very happy to have been given the chance to review it.

1) The paradigm of ruinous solidarity works well especially in light of the qualitative data presented here. However ruinous solidarity will require a level of unselfishness that I am not sure human beings are capable of. As much as I agree with the policy recommendations, I fear that decolonisation will continue to remain a buzz word unless funding structures change as you rightly point.

2) I have some minor comments: whenever abbreviations are introduced please spell them out in full such as IDC, CDC, NIH etc. Also the IDC system needs more clarification - what is the 63% and why is it at this rate? Are the Universities in need of this money from grants because of the systematic reduction of grants from the government [as has happened with public Universities in Europe?] For readers not familiar with the system this description would be helpful.

3) While everywhere else in the manuscript the authors mention elite University in GN, there's one place where the institution is mentioned by one of the respondents - might be good to redact it unless they want to keep it.

4)There's one place where the sentence needs clarification - "Given the history of Global Health, this is a logical and fruitful departure point for change. In a field grounded in coloniality [4] and productive of vast epistemic injustice [5], those who have been cast to the margins of the field, are closest to manifestations of health inequity, and have the most intimate relation with other epistemologies and ways of being should be privileged actors in efforts changing unequal political dynamics. However, this earnest vision of responsibility is one that departs from particular ideas of power. Reflecting broader trends in the field, critiques of ideological power—who shapes thinking and who’s knowledge is taken up—and symbolic power—tacit modes of Western domination reproduced through habits of doing Global Health

work—disproportionately shape how this responsibility is conceptualized. That is, departing from the perspectives, the field can be changed through shifting thinking, desires, and actions, and power can be democratized between North and South when ideologies, knowledges, and

habits change. As anthropologists and other critical scholars have revealed". Here what does the particular ideas of power mean? Is it the previous sentence or the sentence that follows it? Because both are critiques. I found this bit a little confusing.

5) While this is not part of your argument, it might be good to mention that decolonial approaches often are unable to solve the problem of the north within the south. Even when money trickles from GN institutions, elite actors in GS institutions [and elite institutions] end up reaping the benefits.

6. PLOS authors have the option to publish the peer review history of their article (what does this mean? ). If published, this will include your full peer review and any attached files.

**Do you want your identity to be public for this peer review?** For information about this choice, including consent withdrawal, please see our Privacy Policy .

Reviewer #1: No

Reviewer #2: No

Reviewer #3: **Yes: ** Dr Sreeparna Chattopadhyay

---

## [Decision Letter · Decision Letter 1]

26 Feb 2025

PGPH-D-24-01739R1

Soft Money, Hard Power: Mapping the Material Contingencies of Change in Global Health Academic Structures

Dear Dr. Krugman,

Thank you for submitting your manuscript to PLOS Global Public Health. After careful consideration, we feel that it has merit but does not fully meet PLOS Global Public Health’s publication criteria as it currently stands. Therefore, we invite you to submit a revised version of the manuscript that addresses the points raised during the review process.

Please make the minor revisions necessary to address Reviewer 1s comments on abbreviations as noted below.

We look forward to receiving your revised manuscript.

Kind regards,

Emma Campbell, Ph.D

Staff Editor

On behalf of 

Andreas K Demetriades, MBBChir, MPhil, FRCSEd, FEBNS.

Academic Editor

Journal Requirements:

Additional Editor Comments (if provided):

Dear authors

Thank you for submitting a revised version of your manuscript.

The peer review process ha snow recommended publication.

Congratulations for your contribution on this important topic.

Reviewers' comments:

Reviewer's Responses to Questions

**Comments to the Author**

1. If the authors have adequately addressed your comments raised in a previous round of review and you feel that this manuscript is now acceptable for publication, you may indicate that here to bypass the “Comments to the Author” section, enter your conflict of interest statement in the “Confidential to Editor” section, and submit your "Accept" recommendation.

Reviewer #1: All comments have been addressed

Reviewer #3: All comments have been addressed

2. Does this manuscript meet PLOS Global Public Health’s publication criteria ? Is the manuscript technically sound, and do the data support the conclusions? The manuscript must describe methodologically and ethically rigorous research with conclusions that are appropriately drawn based on the data presented.

Reviewer #1: Yes

Reviewer #3: Yes

3. Has the statistical analysis been performed appropriately and rigorously?

Reviewer #1: N/A

Reviewer #3: N/A

4. Have the authors made all data underlying the findings in their manuscript fully available (please refer to the Data Availability Statement at the start of the manuscript PDF file)?

Reviewer #1: Yes

Reviewer #3: Yes

5. Is the manuscript presented in an intelligible fashion and written in standard English?

Reviewer #1: Yes

Reviewer #3: Yes

6. Review Comments to the Author

Reviewer #1: The authors have addressed all comments adequately. My only suggestion would be to revise the abbreviations to make sure that they are spelled out in full form in the first instance that they are mentioned (for example, I think IDCs are mentioned but only spelled out fully in the second instance).

Reviewer #3: I am satisfied with the changes made to the manuscript and addressing the issues that Reviewer #1 and #3 have laid out Reviewer #2 is a different problem and I don't think their comments are fair. The final editorial decision regarding the publication of the manuscript should take this into consideration because the work of these authors are important and needs to be shared with the public. .

7. PLOS authors have the option to publish the peer review history of their article (what does this mean? ). If published, this will include your full peer review and any attached files.

**Do you want your identity to be public for this peer review?** For information about this choice, including consent withdrawal, please see our Privacy Policy .

Reviewer #1: No

Reviewer #3: **Yes: ** Sreeparna Chattopadhyay

---

## [Editor Report · Decision Letter 2]

14 Mar 2025

PGPH-D-24-01739R2

Soft Money, Hard Power: Mapping the Material Contingencies of Change in Global Health Academic Structures

Dear authors,

Thank you for submitting your manuscript to PLOS Global Public Health. After careful consideration, we feel that it has merit but does not fully meet PLOS Global Public Health’s publication criteria as it currently stands. Therefore, we invite you to submit a revised version of the manuscript that addresses the points raised during the review process.

We look forward to receiving your revised manuscript.

Kind regards,

Andreas K Demetriades, MBBChir, MPhil, FRCSEd, FEBNS.

Academic Editor

Journal Requirements:

Additional Editor Comments (if provided):

Dear authors

The peer review process is favourable.

There are a couple of things that lead us to the decision for a further minor revision please:

1. It appears that the methodology may be rather limited and it is not so clear how replicable this study would be with the current reporting.

Please see the relevant publication criteria (https://journals.plos.org/globalpublichealth/s/criteria-for-publication#loc-3); it is recommended that the methodology is described in sufficient detail so as to support a replication of the study.

With this in mind, we would recommend that in your revision you provide additional detail in their methodology.

In addition, and as part of this, please provide- as a supplementary file- a copy of the interview guide that was used to collect the data.

2. In terms of the research question, the results may or may not be generalisable to other settings or institutions.

Since this has not been discussed, please add a few lines about it in a limitations section within the discussion so as to acknowledge the limitations of the study design in regards to the research question, specifically noting the potential issues with generalisability.
---

## [Editor Report · Decision Letter 3]

3 Apr 2025

PGPH-D-24-01739R3

Soft Money, Hard Power: Mapping the Material Contingencies of Change in Global Health Academic Structures

Dear Dr. Krugman,

Thank you for submitting your manuscript to PLOS Global Public Health. After careful consideration, we feel that it has merit but does not fully meet PLOS Global Public Health’s publication criteria as it currently stands. Therefore, we invite you to submit a revised version of the manuscript that addresses the points raised during the review process.

We look forward to receiving your revised manuscript.

Kind regards,

Andreas K Demetriades, MBBChir, MPhil, FRCSEd, FEBNS.

Academic Editor

Journal Requirements:

Additional Editor Comments (if provided):

the most recent "Response to Reviewers" document provided by the authors is blank.

Hence the decision is repeated: Major Revision decision.

Can we please request that the authors provide an updated "Response to Reviewers" document for our consideration, before rendering a final decision or sending it out for further review.
---

## [Editor Report · Decision Letter 4]

10 Apr 2025

PGPH-D-24-01739R4

Soft Money, Hard Power: Mapping the Material Contingencies of Change in Global Health Academic Structures

Dear Dr. Krugman,

Thank you for submitting your manuscript to PLOS Global Public Health. After careful consideration, we feel that it has merit but does not fully meet PLOS Global Public Health’s publication criteria as it currently stands. Therefore, we invite you to submit a revised version of the manuscript that addresses the points raised during the review process.

**Comments from PLOS Editorial Office:**

Thank you for your response to our request for further information regarding methodological details of your study. At present the level of detail provided in the methods section means that it is not clear what you asked your participants.

As noted previously, one of PLOS Global Public Heath's publication criteria is that all studies should be reported in enough detail to allow for the study to be reproduced. Although we acknowledge your comments about the flexible nature of qualitative research, without specifying the details of your methods, it is not possible for a reader or reviewer assess your methods or to design a study that builds upon your research by, for example, asking similar questions to other groups of people in other contexts.

We respectfully disagree that qualitative research is not reproducible (see our guidelines on reporting qualitative research here: https://journals.plos.org/globalpublichealth/s/submission-guidelines#loc-qualitative-research).

Please could you revise your methods section to provide a description of the types of question asked in the interviews. Regarding your interview guide, you state that it contains information that identifies the institution where the study was conducted. Is it possible to provide a de-identified version of the interview guide?

Please submit your revised manuscript by . If you will need more time than this to complete your revisions, please reply to this message or contact the journal office at globalpubhealth@plos.org. Please include the following items when submitting your revised manuscript:

We look forward to receiving your revised manuscript.

Kind regards,

Steve Zimmerman, PhD

PLOS Staff Editor

on behalf of

Andreas K Demetriades, MBBChir, MPhil, FRCSEd, FEBNS.

Academic Editor
---

## [Editor Report · Decision Letter 5]

22 Apr 2025

Soft Money, Hard Power: Mapping the Material Contingencies of Change in Global Health Academic Structures

PGPH-D-24-01739R5

Dear authors

We are pleased to inform you that your manuscript 'Soft Money, Hard Power: Mapping the Material Contingencies of Change in Global Health Academic Structures' has been provisionally accepted for publication in PLOS Global Public Health.

Best regards,

Andreas K Demetriades, MBBChir, MPhil, FRCSEd, FEBNS.

Academic Editor